

# A field-based investigation of behavioural interactions between invasive green crab (*Carcinus maenas*), rock crab (*Cancer irroratus*), and American lobster (*Homarus americanus*) in southern Newfoundland

Nicola Zargarpour[1,2], Cynthia H. McKenzie[2,3] and Brett Favaro[1,2]

[1] Centre for Sustainable Aquatic Resources, Fisheries and Marine Institute of Memorial University of Newfoundland, St. John's, Newfoundland and Labrador, Canada
[2] Department of Ocean Sciences, Memorial University of Newfoundland, St. John's, Newfoundland and Labrador, Canada
[3] Northwest Atlantic Fisheries Centre, Fisheries and Oceans Canada, St. John's, Newfoundland, Canada

Corresponding author
Nicola Zargarpour, 9nz7@queensu.ca

## ABSTRACT

Marine species invasions pose a global threat to native biodiversity and commercial fisheries. The European green crab (*Carcinus maenas*) is one of the most successful marine invaders worldwide and has, in the last decade, invaded the southern and western coastal waters of the island of Newfoundland, Newfoundland and Labrador (NL), Canada. Impacts of green crab on the American lobster (*Homarus americanus*), which are native to Newfoundland, are not well understood, particularly for interactions around deployed fishing gear. Declines in lobster catch rates in invaded systems (i.e., Placentia Bay, NL), have prompted concerns among lobster fishers that green crab are interfering with lobster catch. Here, we conducted a field experiment in a recently-invaded bay (2013) in which we deployed lobster traps pre-stocked with green crab, native rock crab (*Cancer irroratus*) (a procedural control), or empty (control). We compared catch per unit effort across each category, and used underwater cameras to directly observe trap performance *in situ*. In addition, we used SCUBA surveys to determine the correlation between ambient density of lobster and green crab in the ecosystem and the catch processes of lobster in traps. We found: (1) Regardless of the species of crab stocked, crab presence reduced the total number of lobster that attempted to enter the trap, and also reduced entry success rate, (2) lobster consumed green crab, rock crab and other lobster inside traps and (3) there was a positive association between lobster catch and ambient lobster density. Our results suggest that while there was a relationship between in-trap crab density and trap catch rates, it was not linked to the non-native/native status of the crab species.

# INTRODUCTION

Species invasions within marine ecosystems can drive ecological change, which can have wide-reaching implications for fisheries (*Albins & Hixon, 2008*; *Green et al., 2012*). The conservation and management of marine resources requires we understand the impact an invasive species can have on ecosystem dynamics. The European green crab (*Carcinus maenas*) is a crustacean species native to North African and European waters (*Williams, 1984*). Owing to their success in invading new ecosystems and the severity of their impacts once invasion occurs, the European green crab is ranked among the 100 'worst alien invasive species' in the world (*Lowe et al., 2000*).

Green crab were first detected outside their native range in the early 19th century in the north-eastern United States, likely transported via ballast or hull fouling (*Behrens Yamada, 2001*). During the years that followed, green crab populations moved northward, reaching the Canadian border in 1951, establishing in the Bay of Fundy and south-eastern Nova Scotia by the 1960s, and subsequently in north-eastern Nova Scotia by the 1990s (*Carlton & Cohen, 2003*). The first observations of green crab in the province of NL occurred in 2007. Demographic data and surveys suggest that populations have existed for longer, and may have been introduced in the early 2000s (*Blakeslee et al., 2010*). They have since become established across the south and west coasts of the island of Newfoundland (*Best, McKenzie & Couturier, 2017*).

The ecological impacts of green crab invasions can be severe. On the east coast of North America, the species has contributed to the degradation of eelgrass beds (*Garbary et al., 2014*; *Matheson et al., 2016*), declines of soft-shelled clam (*Mya arenaria*) populations (*Tan & Beal, 2015*), as well as having negative impacts on oyster (*Crassostrea virginica*) and mussel (*Mytilus edulis*) beds (*DeGraaf & Tyrrell, 2004*; *Miron et al., 2005*). The crabs' ability to alter ecosystem dynamics make them a serious threat to many crustacean and bivalve fisheries (*DFO, 2011*).

There is concern that European green crab may have negatively impacted lobster populations through predation, competition and habitat modification in Placentia Bay, NL where they were first detected (*DFO, 2016*); and that their continued spread could be detrimental to other areas, particularly the south coast of Newfoundland, where lobster fishing is commercially important. Since 1992, lobster landings in Placentia Bay have severely dropped from 427 t to 20 t in 2017, representing an 80% decline in an area that was once the most productive lobster fishing area in the region (*DFO, 2018*). While the decline began before the onset of a green crab invasion, the presence of the species may complicate lobster recovery in the region. In 2018 during a targeted mitigation in five locations in Placentia Bay 90 tons of green crab were removed and destroyed (K Best, pers. comm., 2018). The ongoing expansion of green crab into productive lobster fishing grounds has raised concern that additional damage to the lobster stock may occur (*DFO, 2016*).

Green crab can impact lobster populations through a variety of mechanisms. For example, *Rossong et al. (2011)*, found juvenile lobster (25–51 mm carapace length) in aquariums that contained green crab favoured shelter use over feeding. This behavioural

impact, which was absent in aquariums without green crab, resulted in reduced energy intake by the lobster. Other laboratory studies have indicated that adult green crab (53–76 mm carapace width) predate on juvenile lobster (28–57 mm carapace length) (*Rossong et al., 2006*), and outcompete sub-adult lobster (55–70 mm carapace length) for a limited food source (*Williams, Floyd & Rossong, 2006*). However, we are unaware of any studies to date examining how these species interact outside the laboratory environment.

It is not clear whether observed declines in lobster catch rates in invaded ecosystems were due to a reduction in lobster abundance or because green crab were interfering with the capture process itself (or a combination of these factors). In this study, we conducted a series of field experiments to determine whether green crab had a direct impact on the ability of commercial lobster traps to catch lobster, and to identify the mechanism by which this impact may be occurring. We employed three pre-stocking conditions (traps with green crab, traps with native crab, and traps with no pre-stocked crab) to assess the impact that green crab had on lobster catch rates. Further, we used a custom-built underwater video camera (described in *Bergshoeff et al., 2017*) to directly observe traps during deployments and assess the behaviour of lobster in relation to both species of crabs. Finally, we conducted SCUBA surveys around the deployed camera traps to assess the extent to which our observations were mediated by the densities of lobster, green crab, and native rock crab in the vicinity of deployed gear.

We tested four non-exclusive questions. First, do traps pre-stocked with green crab catch fewer lobster per deployment than unstocked traps? Second, if green crab do reduce lobster catch rates, do they do so by depleting bait, by physically preventing lobster entry into traps (i.e., blocking entrances), or by inducing lobster to exit the traps? Third, does the density of green crab in the vicinity of deployed traps reduce lobster catch rates? Finally, are these effects specific to green crab, or do they occur with native rock crab as well?

## MATERIALS & METHODS

### Specifications of camera apparatus

To record underwater video of lobster traps fishing *in situ* we designed and assembled a camera system capable of recording full high-definition videos for 13 continuous hours at high resolution (1080p) (see *Bergshoeff et al., 2017*). We mounted the camera housing to a wooden frame constructed around standard commercial single-parlour wire-mesh (3.8 cm mesh size) lobster traps capable of catching all types of crustaceans. We secured the housing to this frame and oriented the camera 58 cm above the trap pointing downward, giving a top-down field of view (FOV) of approximately 105 cm by 170 cm underwater. Using this setup we were able to observe lobster entering, exiting, inside the trap, as well as those around the trap for 13 continuous hours. We attached a Data Storage Tag (DST) to record the depth at which each trap was deployed, but due to a technological malfunction these data were not recoverable. We deployed the traps at depths shallow enough for ambient light to illuminate the pot during the day, therefore we did not use external lights. Traps were deployed within depths reflective of where commercial lobster traps are set (i.e., <20 m at low tide; *DFO, 2016*).

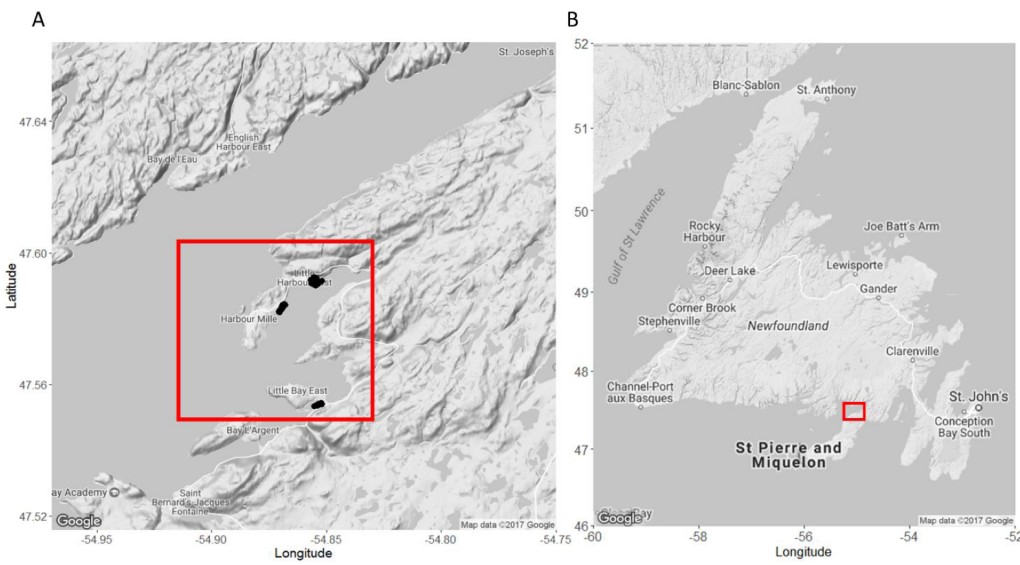

**Figure 1** **Map of our study sites in Little Harbour East and Little Bay East, Fortune Bay, off the southern coast of Newfoundland.** (A) Study sites in Little Harbour East and Little Bay East, Fortune Bay. (B) Our study sites relative to the rest of Newfoundland. Black points indicate where we deployed our traps. Red squares indicate the location of our study sites relative to the rest of Newfoundland. Map A and B imagery ©2017 Google.

## Study site and fieldwork

We conducted field work in nearshore lobster fishing areas around Little Harbour East and Little Bay East, Fortune Bay on the southern coast of Newfoundland (Fig. 1) for five weeks between July–September 2016, directly following the closure of the two-month lobster fishery in Lobster Fishing Area (LFA) 11 (*DFO, 2016*). This allowed us to carry out our field experiments and SCUBA surveys without interfering with local fishing operations or having to work around other fishing gears and vessels. We selected specific sites within the region based on advice from fishers, scientists (K Matheson, Aquatic Science Biologist, Fisheries and Oceans Canada (DFO), pers. comm., June 2016), and local members of the community.

We deployed all traps at depths ranging from 3 to 19 m (mean $\pm$ 1 S.D. $= 9.52 \pm 4.27$ m as measured at low tide), using a 4.2 m rigid-hull inflatable boat, and we recorded GPS coordinates of all traps as they were set. Traps were batched into deployments consisting of three camera-equipped traps (control, pre-stocked with green crab, or pre-stocked with rock crab: see below) and two traps without cameras. Every trap was baited before each deployment with equal amounts of frozen herring (about one half of a large fish) in bait bags tied inside the "kitchen" (a gear-specific term for the entrance compartment, Fig. 2A). Soak durations (i.e., the time between deployment and retrieval) were approximately 24 h and traps were set ~50 m apart horizontally. No other fishing activity occurred in the vicinity of our experiment. Upon retrieving each trap, we recorded the size (carapace length- tip to tip) and sex of every lobster captured. All lobster were released where caught. We counted the number of crabs caught in the trap, and retained the green crab to be

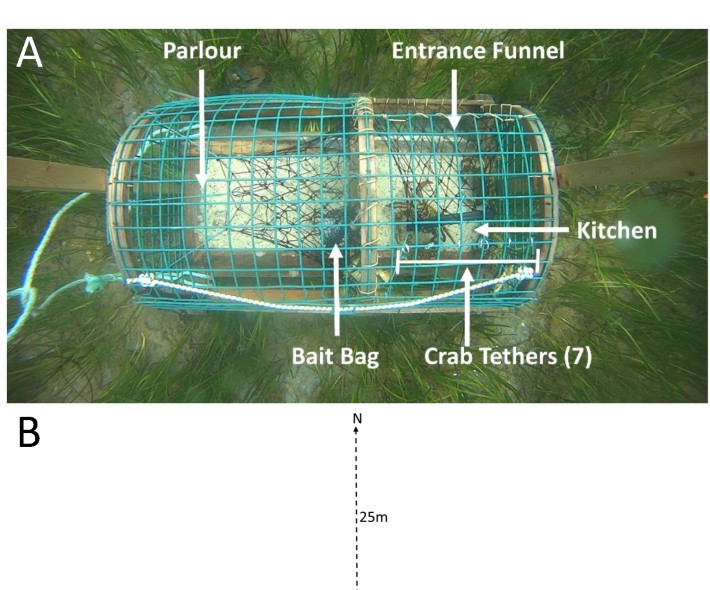

**Figure 2 Top-down view of a green crab pre-stocked camera trap and SCUBA transects conducted on camera traps.** (A) The mesh entrance funnel leads into the kitchen compartment where the bait bag is stored and where 7 crabs were tethered using fishing line tied to split metal rings that were attached to wire cells on the top of the trap. The parlour refers to the area where lobsters are retained. (B) Dotted lines indicate the 25 m transects, oriented North, South, West and East of the camera trap (shown in the centre of the image). The 2 m marking indicates the area in which divers recorded counts of green crab, lobster and rock crab.

euthanized on shore in accordance with Memorial University's animal care protocol. All traps were re-baited and re-deployed within the study area. In total, we conducted 17 deployments (with five traps per deployment, total $n = 85$).

The project was approved as a 'Category A' study by the Institutional Animal Care Committee at Memorial University as it involved invertebrates (project # 15-02-BF), and all field research was conducted under experimental license NL-3271-16 issued by Fisheries and Oceans Canada.

## Pre-stocking field experiment

To determine the influence of crabs (both native and invasive) inside the trap on subsequent lobster entries, we 'pre-stocked' two of the three camera-equipped traps with either seven native rock crabs (*Cancer irroratus*) or seven invasive green crabs (*Carcinus maenas*) tethered inside the kitchen. The density of pre-stocked crabs (seven per trap) was selected

to impose a strong effect of crabs (green crab and rock crab) entering the traps in volumes, replicating conditions observed in previous field studies at sites with higher densities of green crab than those observed in our study area. The technique of tethering has been described in previous studies, and is commonly used to evaluate behavioural interactions among species (*Wahle & Steneck, 1992*; *Watson & Jury, 2013*). We tethered the crabs using fine fishing line looped around the rear two legs of the crab and tied (leash-like) to split metal rings attached to the wire cells on the top of the kitchen (Fig. 2A). This allowed crabs to move freely around the kitchen without being able to exit the trap. The third camera trap served as a control and was not pre-stocked with any crabs. We captured the green crab used for pre-stocking with Fukui traps in intertidal areas bordering the sites where we deployed our lobster traps. We collected rock crab via SCUBA from areas adjacent to deployment sites. All tethered crabs (both green crab and rock crab) had carapace widths >5 cm (measured point to point). Each crab was used in only a single experimental trial, and was either destroyed (green crab) or returned to the site from which they were collected (rock crab) on conclusion of the trial.

## SCUBA surveys

Due to logistical limitations, we were only able conduct dive surveys for 11 of 17 deployments (one dive per trap, $n$ dives = 33). Within each dive, divers recorded the depth at which the trap was deployed, and collected data along four transects oriented North, South, West and East of each trap (Fig. 2B).

For each survey, two SCUBA divers descended on camera traps. The divers conducted the survey swimming side-by-side, where one diver reeled out the transect line, orienting in one of the four cardinal directions, while the other diver swam alongside simultaneously and recorded counts of lobster, green crab and rock crab within two metres of the transect line up to the end of the transect at 25 m. After 25 m the divers swam back to the trap and reeled in the transect tape. This process was repeated, orienting in the remaining three directions. We repeated this for each camera trap, for a total surveyed area of 200 $m^2$ per trap, and 600 $m^2$ per deployment.

## Video analysis

We collected approximately 663 h of underwater video footage across 17 deployments. Following the field study, we scored the videos manually, following protocols described in previous literature (*Favaro et al., 2012*; *Jury et al., 2001*; *Meintzer, Walsh & Favaro, 2017*). Specifically, we recorded the following quantitative parameters for lobster, rock crab and green crab: (1) the number, direction and duration of entry attempts as well as the proportion of those entries that were successful versus failed, (2) the number, direction and duration of exits from the trap, (3) the time spent feeding on the bait, (4) the number and duration of interspecific aggression events, and (5) the number and duration of predation events.

We defined an entry attempt as an instance where more than half the individual's body length crossed into the trap. The result and duration of each attempt was scored as either failed, where the individual retreated outside the trap, or as a success, where

the individual's body crosses entirely into the trap. Individuals could enter/exit the trap through the entrance, through the wire cells in the trap kitchen and parlour (Fig. 2A), or through the escape slats in the top and bottom of the trap. Note, each entry attempt was considered 'new' given that individuals were not individually identifiable, thus we assume that some individuals were counted multiple times.

We defined interspecific aggression as agonistic behaviours involving individuals engaging in threat displays including a meral spread (chelae raised and held laterally, outwards from the body), or any physical contact including touching and grasping actions (*Haarr & Rochette, 2012*; *Rossong et al., 2006*). We defined predation as behaviours involving one individual being consumed by another. For each predation event observed, we visually estimated the claw size of the predator(s) relative to a fixed point on the trap of known length (3.8 cm wire cells) using Adobe Photoshop. We assigned lobsters into bins of small, medium, and large, with estimated claw sizes being below 6 cm, between 6 cm and 8 cm, or above 8 cm, respectively.

## Statistical analysis—catch data

In this paper we use the software package R (*R Core Team, 2015*). We carried out data exploration in accordance with the protocol described in *Zuur, Ieno & Elphick (2010)*.

To model lobster catch as a function of the covariates a Poisson generalized linear mixed-effects model (GLMM) with a log link function was used. GLMMs can handle data that violates assumptions required for simple linear models and are therefore a common technique for analyzing ecological data (*Zuur et al., 2009*). The Poisson distribution is typically used for count data and the log link function ensures positive fitted values, and if the Poisson GLMM is overdispersed, a negative binomial model can be used instead (*Zuur & Ieno, 2016*). Fixed covariates in our full model are trap pre-stocking condition (categorical, four levels: unstocked, green crab pre-stock, rock crab pre-stock, no camera/unstocked) and soak duration (continuous) [Eq. (1)]. We included an interaction term between trap pre-stocking condition and soak duration. We included deployment number (1–17) as a random effect. As we only had fine-scale depth data for 11/17 deployments, we could not include it as a covariate, so we created exploratory plots to assess whether relationships existed between lobster catch and deployment depth (Fig. S1).

We then conducted stepwise model simplification, sequentially dropping non-significant terms until all terms in the model were statistically significant (procedure outlined in *Crawley, 2012*). This procedure was used for all models presented in this paper, and we used the lme4 package (*Bates et al., 2017*) to fit models. We verified model assumptions by plotting residuals against fitted values. Residuals met assumptions for normality, homogeneity and independence, and there was no evidence of overdispersion. We interpreted reduced models.

Lobster Catch$_{ij}$ ~ Poisson($\mu_{ij}$)

$E$(Lobster Catch$_{ij}$) $= \mu_{ij}$

$\log(\mu_{ij}) =$ Trap Prestocking Condition$_{ij}$ + Soak Duration$_{ij}$

+Trap Prestocking Condition$_{ij}$ × Soak Duration$_{ij}$ + Deployment$_i$

$$\text{Deployment}_i \sim N(0, \sigma^2). \tag{1}$$

We used a linear mixed effects model using carapace length as our response variable, with fixed covariates of prestocking condition, soak duration, and a prestocking by soak duration interaction, with deployment as a random effect [full model; Eq. (2)].

$$\text{Carapace Length}_{ij} \sim N(\mu_{ij}, \sigma^2)$$

$$E(\text{Carapace Length}_{ij}) = \mu_{ij}$$

$$\text{Var}(\text{Carapace Length}_{ij}) = \sigma^2$$

$$\text{Carapace Length}_{ij} = \text{Trap Prestocking Condition}_{ij} + \text{Soak Duration}_{ij}$$

$$+ \text{Trap Prestocking Condition}_{ij} \times \text{Soak Duration}_{ij} + \text{Deployment}_i$$

$$\text{Deployment}_i \sim N(0, \sigma^2). \tag{2}$$

## Statistical analysis—video data

We used a negative binomial GLMM to test the fixed effect of trap pre-stocking condition on lobster entry time [Eq. (3)], and incorporated deployment as a random effect. The distribution of our lobster entry time data was best explained by a negative binomial distribution.

$$\text{Entry Time}_{ij} \sim NB(\mu_{ij}, \text{theta})$$

$$E(\text{Entry Time}_{ij}) = \mu_{ij}$$

$$\text{Var}(\text{Entry Time}_{ij}) = \mu_i + (\mu_i^2/\text{theta})$$

$$\log(\mu_{ij}) = \text{Trap Prestocking Condition}_{ij} + \text{Deployment}_i$$

$$\text{Deployment}_i \sim N(0, \sigma^2). \tag{3}$$

To examine whether there was an association between trap pre-stocking condition and the proportion of successful versus failed entry attempts by lobster, we used the prop.test function in R which uses Pearson's chi-squared test statistic to test the null that the proportions (probabilities of success) in several groups are the same (*R Core Team, 2015*).

## Statistical analysis—SCUBA data

The SCUBA-based transect survey component of our study provided a context for what we observed in our catch and video data and allowed us to obtain estimates of lobster density that were temporally matched to the video and trap catch data. We used a GLMM to measure the impact of the ambient lobster density (continuous) and trap pre-stocking condition (categorical, four levels) on lobster catch, and an interaction between pre-stocking condition and density [Eq. (4)].

The distribution of our catch data was best explained by a Poisson distribution. We simplified the model using stepwise removal of non-significant terms (*Crawley, 2012*) and fit the model using the lme4 package (*Bates et al., 2017*). We plotted residuals versus fitted values to verify the model assumptions. Residuals met the assumptions for normality, homogeneity and independence, and there was no evidence of overdispersion.

$$\text{Lobster Catch}_{ij} \sim \text{Poisson}(\mu_{ij})$$

$$E(\text{Lobster Catch}_{ij}) = \mu_{ij}$$

$$\log(\mu_{ij}) = \text{Trap Prestocking Condition}_{ij} + \text{Ambient Lobster}_{ij}$$

$$+ \text{Trap Prestocking Condition}_{ij} \times \text{Ambient Lobster}_{ij} + \text{Deployment}_i$$

$$\text{Deployment}_i \sim N(0, \sigma^2). \tag{4}$$

To examine whether there was an association between trap pre-stocking condition and the proportion of successful versus failed entry attempts by lobster for deployments where video data and SCUBA data were temporally matched, we used the prop.test function in R.

## RESULTS

### Catch data

Across 85 individual trap deployments soak durations ranged between 15.2 and 26.6 h (mean ± 1 SD = 21.9 ± 3.1). We did not detect a relationship between soak duration and the number of lobster caught across pre-stocking conditions (Fig. S2). Similarly, we found no relationship between depth and either lobster catch or carapace length (Figs. S1 and S3).

We caught a total of 326 lobsters, six green crabs, and three rock crabs across the entire study. Traps pre-stocked with green crab caught between 47% fewer, and 1.97% more lobster than unstocked traps (95% C.I.; $\beta = -0.312$, S.E. = 0.168, $p = 0.064$; Fig. 3A; Table 1). Traps pre-stocked with rock crab caught between 54% and 8.9% fewer lobster than unstocked traps, (95% C.I.; $\beta = -0.436$, S.E. = 0.175, $p = 0.013$; Fig. 3A; Table 1).

We found no significant impact of the presence of the camera apparatus on lobster catch (GLMM: No camera, $\beta = -0.225$, S.E. = 0.140, $z = -1.610$, $p = 0.107$; Fig. 3A; Table 1). Soak duration did not significantly influence this relationship and was removed from the model via stepwise reduction. Model validation indicated our choice of model was appropriate through visual inspection of residuals. The model had a dispersion parameter of 1.1, indicating it was not overdispersed.

Lobster size (carapace length) ranged from 48 to 98 mm (mean ± 1 SD = 80.4 ± 6.1 mm). The average size of lobster caught did not vary substantially across trap pre-stocking condition and 70.9% of the lobster captured were of sub-legal size (<82.5 mm) (Fig. 3B).

### Video data

Of the 663 h of underwater video footage collected across 17 deployments, 452 h had adequate ambient lighting for analysis. We analyzed the same amount of video (~150 h) for each trap pre-stocking condition.

Across all videos we observed a total of 3,801 lobster entry attempts across all traps. There were 1,625 entry attempts in unstocked traps, 1,119 in traps pre-stocked with green crab, and 1,057 in traps pre-stocked with rock crab. The proportion of these attempts that resulted in successful entry differed across pre-stocking condition (unstocked: 69.7%, green crab: 45.3%, rock crab: 49.8%). We observed 1,070 lobsters exiting from unstocked traps, 409 exiting from traps stocked with green crab, and 494 exiting from traps stocked with rock crab (Fig. 4). Because we could not identify individual lobsters, we tracked raw

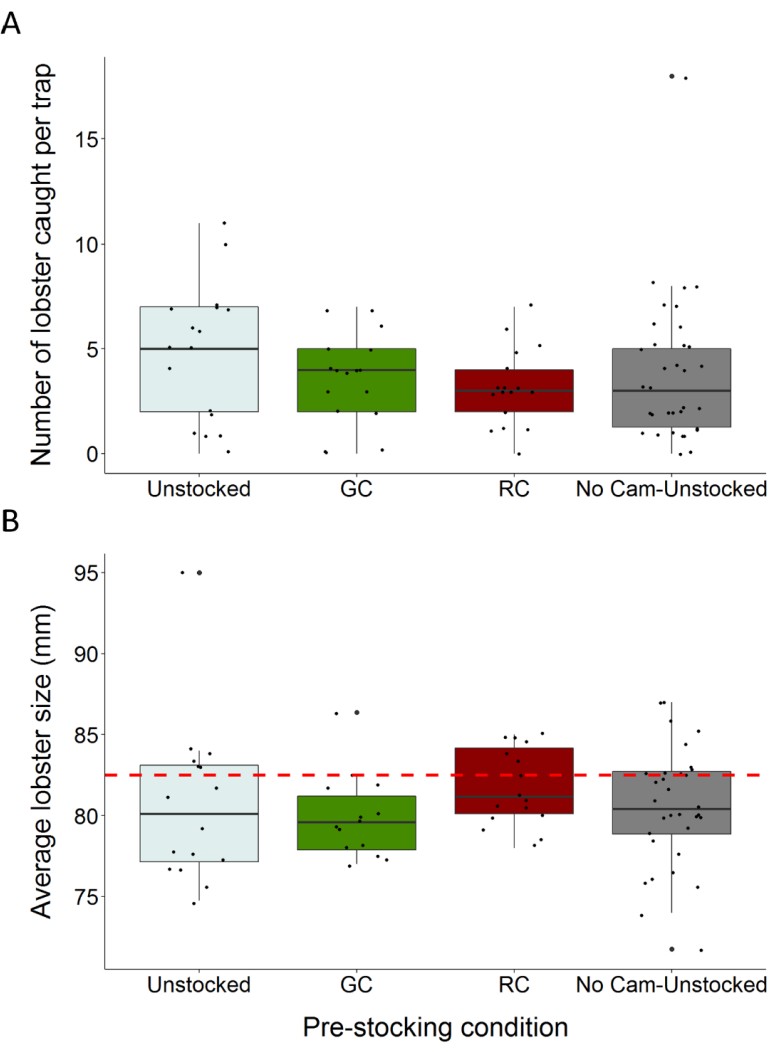

**Figure 3 Boxplots comparing total lobster catch and average lobster size to trap pre-stocking condition.** (A) The distributions of total lobster catch according to trap pre-stocking condition (B) the distributions of average lobster size (carapace length) across trap pre-stocking conditions (the dotted red line indicates the minimum legal size of 82.5 mm carapace length). Each black dot represents a single observation.

number of entries and exits (e.g., if a lobster did a successful entry attempt, exited, and then re-entered, it would be counted as two entries and one exit).

When we accounted for lobster entry attempts through the entrance only (i.e., excluding small individuals that attempted entries through the wire cells and escape slats) we found 30.1% ($N = 139$) were successful in the unstocked trap, while 17.1% ($N = 106$) and 17.7% ($N = 105$) of entry attempts were successful for green crab and rock crab pre-stocked traps, respectively (Fig. 5).

The proportion of successful entries was higher in the unstocked traps than in either the green crab or rock crab pre-stocked traps for all lobster attempts ($x^2 = 190.072$, $df = 2$, $p < 0.001$; Fig. 6A) as well as for those only through the trap entrance ($x^2 = 36.049$, $df = 2$,

**Table 1** **Model output from Eq. (1). Estimated regression parameters, standard errors, z-values and P-values.**

|  | Estimate | Std. error | z value | P-value |
|---|---|---|---|---|
| Intercept | 1.4476 | 0.1770 | 8.181 | 2.82e−16 |
| Green crab pre-stock | −0.3124 | 0.1684 | −1.855 | 0.0636 |
| Rock crab pre-stock | −0.4364 | 0.1747 | −2.499 | 0.0125 |
| Unstocked trap | −0.2247 | 0.1396 | −1.610 | 0.1074 |

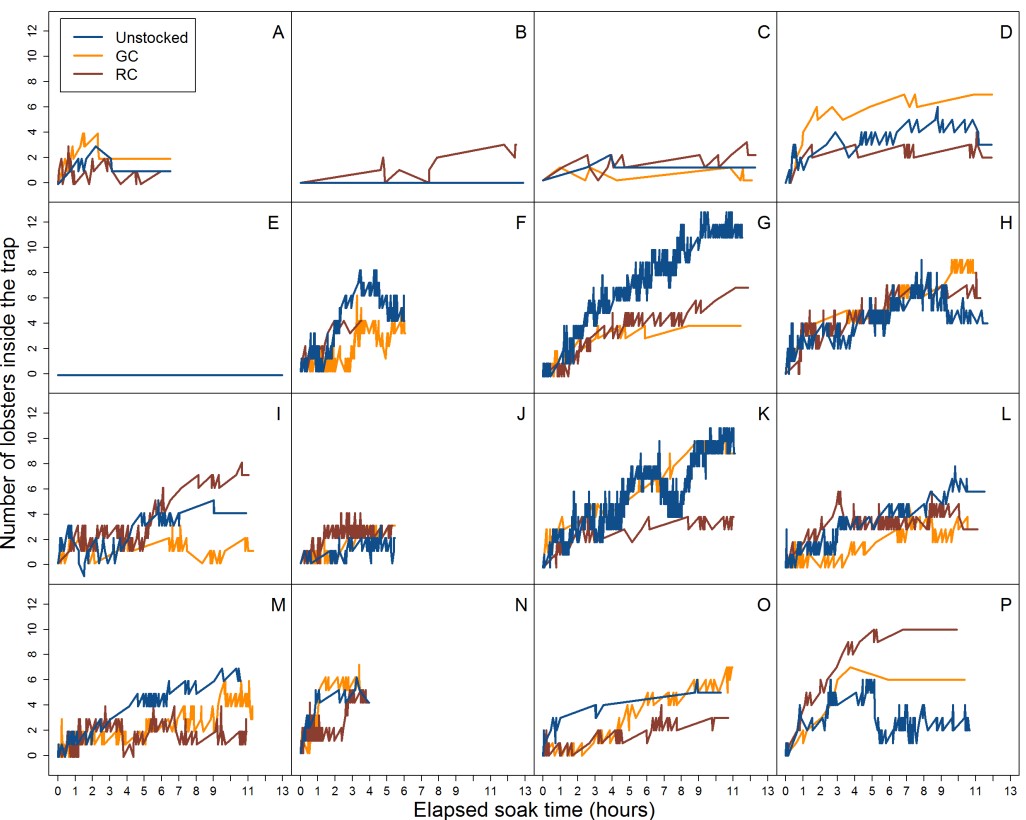

**Figure 4** **Video data analysis of lobster accumulation over the elapsed soak time for all entries and exits (i.e., large and small-bodied lobster).** Each coloured line represents pre-stocking condition of traps. A–P represent the deployment.

$p < 0.001$; Fig. 6B). Smaller lobster were able to easily crawl through the wire cells on the kitchen and parlour sides of the trap, as well as through the escape slats on the top and bottom of the trap. We did not observe any difference in the proportion of successful entries across trap pre-stocking conditions for attempts made by small lobster through the wire cells or escape slats (Fig. 6C). We did not detect any difference in the duration of successful lobster entry attempts through the trap entrances according to trap pre-stocking condition (Fig. S4; Table 2).

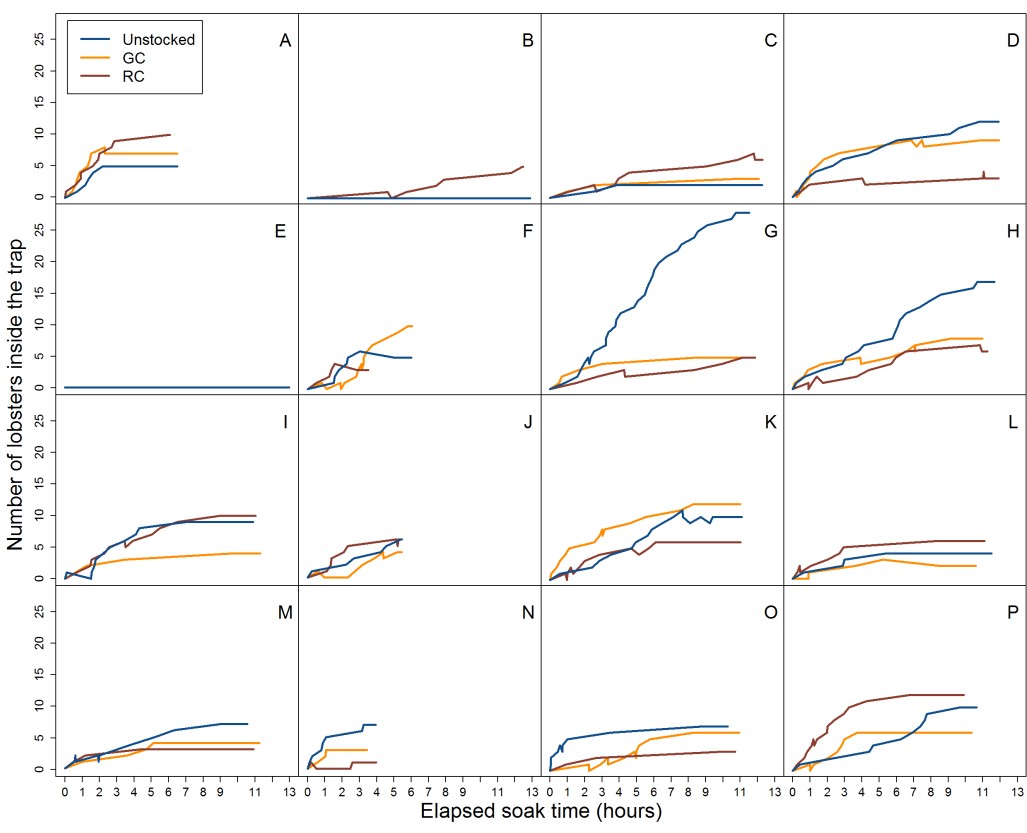

**Figure 5** Video data analysis of lobster accumulation over the elapsed soak time for entries and exits through the trap entrance only (i.e., primarily large-bodied lobster). Each coloured line represents trap pre-stocking condition. A–P represent the deployment.

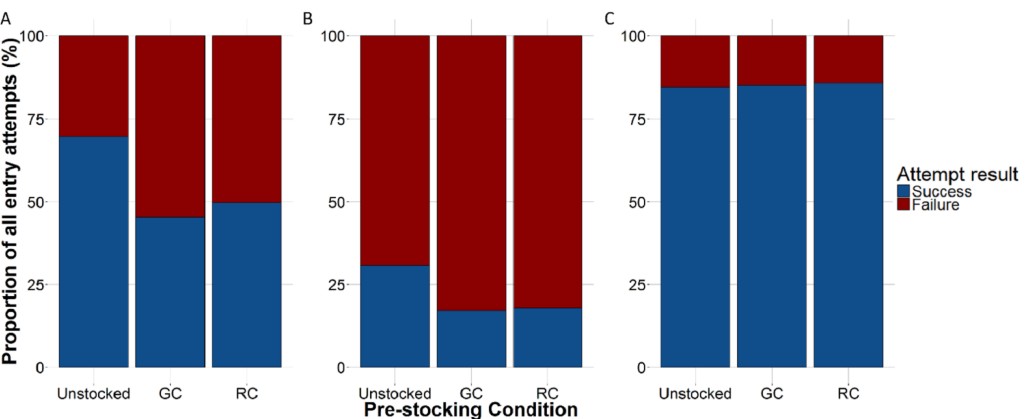

**Figure 6** The proportion of lobster entry attempts that were either successful (blue) or failed (red) according to trap pre-stocking condition. (A) All entry attempts. (B) Entry attempts through the trap entrance only (i.e., primarily large-bodied lobster). (C) Entry attempts through the wire cells or escape slats only (i.e., primarily small-bodied lobsters).

**Table 2  Model output from Eq. (2). Estimated regression parameters, standard errors, z-values and P-values.**

|  | Estimate | Std. error | z value | P-value |
|---|---|---|---|---|
| Intercept | 1.4447 | 0.1916 | 7.541 | 4.66e−14 |
| Green crab pre-stock | −0.2908 | 0.2064 | −1.409 | 0.1589 |
| Rock crab pre-stock | −0.4155 | 0.2118 | −1.962 | 0.0498 |
| Unstocked trap | −0.2278 | 0.1735 | −1.313 | 0.1891 |

**Table 3  Model output from Eq. (3). Estimated regression parameters, standard errors, z-values and P-values.**

|  | Estimate | Std. error | z value | P-value |
|---|---|---|---|---|
| Intercept | 1.1051 | 0.1899 | 5.820 | 5.9e−09 |
| Ambient Lobster | 0.0160 | 0.0071 | 2.258 | 0.024 |

There was little difference between the number of lobster inside traps after a full deployment versus at the end of the video analysis period for each trap pre-stocking condition (mean difference $\pm$ 1 SD = unstocked, 1.2 $\pm$ 2.5; green crab pre-stock, −0.3 $\pm$ 2.9; rock crab pre-stock, −0.5 $\pm$ 2.5; Fig. S5).

We observed very few green crab or rock crab entering traps in any stocking condition. We observed 45 entry attempts made by green crab, of which 27 were successful. Rock crab made 18 entry attempts, of which 17 were successful. Green crab and rock crab were observed exiting the camera traps 19 and 13 times respectively. All these crabs occurred in just two deployments, meaning we observed no crab entry in most videos.

We observed 60 events in which lobster consumed all or part of another organism while in the trap. There were 36 predation events against tethered rock crab, 20 against tethered green crab, and four against other lobster. Lobsters that engaged in predation activities tended to have larger claw sizes than those that did not predate on pre-stocked rock crab, green crab, or lobster (Fig. S6).

## SCUBA surveys

Lobster catch was positively associated with changes in the density of lobster in the vicinity of the trap (GLMM: ambient lobster, $\beta = 0.016$ S.E. = 0.007, $z = 2.258$, $p = 0.024$; Fig. S7; Table 3). Trap pre-stocking condition did not significantly influence this relationship and was removed from the model via stepwise reduction. Model validation indicated our choice of model was appropriate and the model had a dispersion parameter of 0.95, indicating it was not overdispersed. The output from the model is provided in Table 3.

The number of ambient lobster ranged from 5 to 50 (mean $\pm$ SD = 21.97 $\pm$ 11.24), ambient green crab ranged from 0 to 8 (mean $\pm$ SD = 0.27 $\pm$ 1.40), and ambient rock crab ranged from 0 to 9 (mean $\pm$ SD = 1.70 $\pm$ 2.05). Densities of lobster, green crab and rock crab determined by SCUBA surveys ranged from 0.025 to 0.25 lobster/m$^2$, 0 to 0.04 green crab/m-squared, and 0 to 0.045 rock crab/m$^2$.

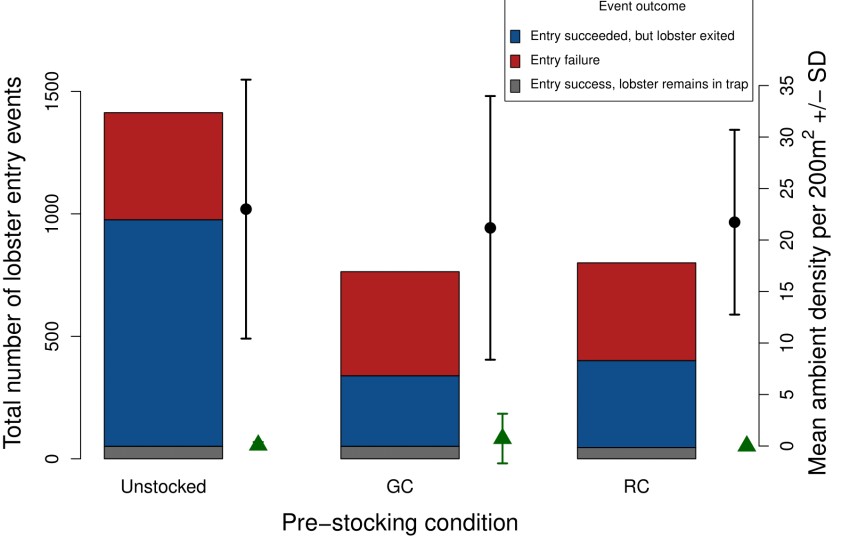

**Figure 7** **Total number of lobster entry events across trap pre-stocking condition for deployments with both video and SCUBA data ($N = 11$).** Event outcomes include entry failure (red); entry succeeded, but lobster exited (blue); entry succeeded, lobster remains in the trap (grey). The black points and lines indicate mean ambient lobster density ±1 SD surveyed around each trap pre-stocking condition. The green triangles and lines indicate mean ambient green crab density ±1 SD surveyed around each trap pre-stocking condition m$^2$.

The average number of ambient lobster did not vary significantly across trap pre-stocking conditions (mean ± 1 SD = unstocked, 23.0 ± 12.57; green crab pre-stock, 21.18 ± 12.80; rock crab pre-stock 21.73 ± 8.97; Fig. 7).

# DISCUSSION

The goal of this study was to determine whether the presence of green crab inside lobster traps and in the vicinity of deployed traps influenced the trap's lobster catch rate. Our study demonstrated that (1) the presence of both green crab and native rock crab impacted the total number of lobster that attempted to enter the trap and the number that successfully entered, (2) lobster can predate on green crab, rock crab and other lobster inside traps and (3) there is a positive association between lobster catch and ambient lobster density.

## Catch data

We found traps pre-stocked with both species of crabs caught fewer lobster than unstocked traps, though the reduction associated with green crab was not technically statistically significant ($p = 0.06$).

The number of green crab observed around our traps via SCUBA surveys were relatively low. As a result, we were not able to robustly assess the relationship between ambient green crab density and lobster catch rates. Since completing this study, the density of green crab in Fortune Bay has increased (C McKenzie, pers. obs., 2018). Repeating this study in an area with a higher density of invaders may yield additional insight.

The sizes of captured lobster in our study were similar across all trap pre-stocking conditions and the majority were of sub-legal size (<82.5 mm). Given that our fieldwork was conducted after closure of the commercial fishery, it is possible that fewer legal sized (>82.5 mm) lobster were available to be captured. Previous studies have found that the presence of larger individuals inside traps can reduce the catch of smaller individuals for lobster (*Watson & Jury, 2013*) as well as for green crab and rock crab (*Miller & Addison, 1995*). *Watson & Jury (2013)* suggest that this may be mediated by bait guarding, particularly when smaller individuals attempt to enter the trap. Given that adult lobster are much larger than the pre-stocked green crab and rock crab, it is perhaps unsurprising that we did not observe a difference in average lobster size across pre-stocking condition from our catch data.

We did not detect a relationship between the duration of time a trap was soaking in the water and the number of lobster caught across trap pre-stocking condition (Fig. 5). Our underwater video camera allowed us to observe and quantify the accumulation of lobster over the first 13 h of trap deployments, after which the camera would turn off and the trap would be left to soak until retrieval the following day. We did not observe more lobster captured at the end of the deployment (24 h soak period) versus at the end of the video observation duration (13 h) for any of the trap pre-stocking conditions. Our results are in general agreement with those of *Miller & Rodger (1996)*, who found traps retrieved more than twice in a 24 h period captured more lobster than those retrieved only once due to trap saturation.

## Video data

The ambient density of green crab was low, and in many videos we saw none at all in this area that was recently-invaded at the time of the study. As a result, we did not find support for the hypotheses that green crab invade the trap and either (1) deplete the bait, (2) impede entry or (3) force lobster to exit the trap (increased escapement).

We found no support for the bait depletion hypothesis—tethered green crab did not deplete the bait in these traps. Chemical signals released by conspecifics, competitors, predators, and mates, as well as internal motivational states (e.g., activity level, moult stage, reproductive status, hunger level), and environmental factors (e.g., flow, temperature, light intensity, photoperiod), may contribute to the sensitivity of green crab to feeding stimulants (*Atema, 1995*; *Atema, 1985*; *Breithaupt & Atema, 2000*; *Hayden et al., 2007*; *Hazlett, 2003*; *Hazlett, Acquistapace & Gherardi, 2006*; *Moore & Howarth, 1996*; *Sneddon et al., 2003*; *Zimmer-Faust, O'Neill & Schar, 1996*; *Zimmer-Faust, 1991*; *Zimmer-Faust, 1989*). Further, a laboratory study conducted by *Fletcher & Hardege (2009)* found that male green crab took significantly longer to respond to food stimuli as a result of agonistic encounters. Given the complexity of chemoreception in its mediatory role of feeding behavior in marine crustaceans, we were unable to investigate the influence of tethering on the feeding response of pre-stocked green crab in the field.

Our finding that the proportion of successful lobster entries were reduced in crab-stocked traps aligns with other pre-stocking (*Richards, Cobb & Fogarty, 1983*) and field studies (*Jury et al., 2001*) that suggest interactions between lobster inside and outside traps can strongly

limit lobster entry and catch. *Jury et al. (2001)* found that only 11% of lobster entry attempts were successful when the kitchen was occupied versus 64% when the kitchen was vacant. In addition, they note that entry attempts may also be influenced by competition outside the trap where both lobster and crab have been observed competing aggressively for trap entry (*Jury et al., 2001*). We did not observe any instances of interactions between untethered crabs and lobster during entry attempts. This is likely due to the low density of crabs around traps as observed via our SCUBA surveys. Interestingly, we did not detect any difference between crab pre-stocked traps and unstocked traps for attempts made by small lobster through wire cells and escape slats. This could be attributed to smaller lobster having more opportunities to enter the trap (through the top and bottom escape slats, and the 3.8 cm wire cells on the kitchen and parlour sides of the trap), whereas larger lobster could only enter through the trap entrance (where they would directly encounter the pre-stocked crabs in the trap kitchen). Small lobster may therefore have been less likely to interact with/directly encounter the pre-stocked crabs as these were tethered in the trap kitchen only.

We found lobster did not take substantially longer to complete a successful entry through the entrance into crab pre-stocked traps compared to unstocked traps. Our video data provided no evidence to suggest that lobster entry was physically impeded by the presence of rock crab or green crab, rather fewer lobster attempted to enter traps where crabs were present. Furthermore, our dive data show that this was not a consequence of lower lobster density around the crab pre-stocked traps.

Results from various laboratory experiments have suggested that the odor of trapped crab and lobster could repel individuals that might otherwise be attracted to either the bait or the trap as a shelter (*Miller, 1978*). It is possible that these factors were occurring but we were not able to detect them from our video and SCUBA analyses.

Importantly, we recorded all videos during daylight hours where ambient light was sufficient to make observations. As a result, we were not able to assess interactions that occurred overnight. While previous research suggests that lobster may be more active during the night, in a preliminary field study conducted in 2015 we did not find a substantial difference in lobster catch between traps deployed during the daytime compared to traps deployed at night. Furthermore, a field study by *Jury et al. (2001)*, found no evidence that lobster will enter traps more often during the night. Our data concur with these findings, as we found our catch data (daytime and nighttime) did not significantly differ from our video data, captured during the initial 13 h of trap deployment (daytime).

Our catch data alone suggested that traps pre-stocked with both species of crabs caught fewer lobster than unstocked traps, though the reduction due to green crab was not statistically significant ($p = 0.06$). Our video data demonstrated that when traps were pre-stocked with crabs, successful entries as well as entry attempts of lobster were significantly reduced. Our trap deployments collated 85 observations of catch data and recorded 452 h of underwater video data which was analysed to produce around 10,000 individual observations. This provides a compelling difference between the breadth of information we can glean from 85 observations of catch data as compared to 452 h of in-depth video analysis. Taken together, the results from our catch and video data, suggest

that while there is a relationship between in-trap crab density and trap effectiveness, it is not linked to the non-native/native status of the crab species. Although, if the number of green crab are significantly higher due to invasion status then it would be more likely that green crab would be the crab in the trap interacting with the lobster.

Previous laboratory studies have demonstrated that predation is possible between lobster and green crab (*Goldstein et al., 2017*). Our study is the first to provide evidence of this dynamic occurring inside lobster traps observed in the field using long duration underwater video. In some instances, pre-stocked crabs were consumed in preference to a full bait bag. It is important to note that crabs in our study were tethered inside traps and were not able to flee from aggressors, potentially forcing these interactions. In general, this dominance dynamic aligns with a large body of literature on agonistic behaviours in crustaceans showing that the larger animal typically dominates in aggressive contests (*Dingle, 1983*; *Hyatt, 1983*; *Rossong et al., 2006*). Interestingly, our underwater video also enabled us to observe multiple events of cannibalism where trapped adult lobster were dismembered and consumed by other lobster in the parlour of the trap. In previous laboratory trials *Haarr & Rochette (2012)* found that at high densities juvenile lobster (20–45 mm carapace length) would consume conspecifics. However, they did not observe juvenile lobster predating on green crab.

## SCUBA surveys

Due to the low ambient density of green crab observed via our SCUBA surveys we were unable to test whether green crab-induced reductions of lobster catch would be more severe where ambient green crab density was higher. Nevertheless, our SCUBA surveys provided us with two pieces of information that contextualize our results. First, there is a positive relationship between lobster catch and the density of lobster around the traps, which is consistent with a field study by *Watson & Jury (2013)*. Second, we did not detect any difference in the mean density of lobster around traps in each pre-stocking condition (Fig. 7), implying that any differences we detected were due to pre-stocking condition and dynamics that occurred within the trap, rather than being a result of different lobster densities in regions in which traps were deployed.

We believe the low ambient densities of green crab observed during our SCUBA surveys represented actual low densities, and were not an artifact of low detection probability.

While detection probability can vary across species (*Tremblay & Smith, 2001*), we restricted dives to daylight hours and our sites were composed of sand and small cobble, which provided a high contrast background against which we were able to detect these species.

## Procedural control

The use of a procedural control (rock crab) was critical to this study. Since lobsters' response to both green crab and rock crab was similar, it demonstrated impacts on lobster catch were not specific to green crab but were rather due to the presence of either crab species in the traps. This finding aligns with those of *Howard, Therriault & Côté (2017)* which found non-native crabs did not reduce prey abundance via direct consumption any

more than native crabs. *Howard, Therriault & Côté (2017)* also reported on the paucity of studies that directly compared impacts of native versus non-native species. Had we not incorporated a procedural control, our conclusion would likely have been that green crab caused declines in trap effectiveness, with the implication that something about their identity as an invader was the root cause. We recommend future studies investigating green crab impacts incorporate a direct comparison with the impacts of native species. In some cases, it may be the nature of the invasion (e.g., rapid population growth) rather than the identity of the species that causes impacts.

Nevertheless, our findings do not imply that rapid increases in the green crab population will be benign to the lobster fishery, or to lobster populations. In heavily invaded systems green crab have been observed readily accessing and being captured by lobster traps (*Goldstein et al., 2017*). In the Great Bay Estuary, New Hampshire, *Goldstein et al. (2017)*, captured ∼8.5 times more green crab than lobster. This implies that reductions in catch rates may not be due to a specific unique quality of the invader but rather the sheer abundance of green crab relative to native species.

## CONCLUSIONS

This study has shown the presence of crabs inside lobster traps can reduce the effectiveness of lobster traps. Crabs in traps cause fewer lobster to attempt entry, and reduce the success rate of entries. This effect was observable with both native and invasive crabs. Using SCUBA survey data we determined the difference could not be explained by differences in ambient lobster density across stocking conditions.

As green crab spread in extent and grow in density around Newfoundland (*DFO, 2016*), it will become increasingly important to understand how the invasion is specifically affecting fishery performance. As the density of green crab increases, there will be more potential for fishery interactions. While the collapse of lobster catch rates in neighbouring Placentia Bay pre-date the green crab invasion (and so cannot be blamed on green crab alone *Best, McKenzie & Couturier, 2017*; *DFO, 2018*), the lack of recovery in this heavily-invaded system may signal a warning to fisheries in the Fortune Bay region—that if green crab continue to grow in abundance, it is likely traps will continue to perform worse. While ecosystem responses to invasion must continue to be studied, future research should also examine fishing gear performance across gradients of invader density to better understand how invaders can impact the capture process itself.

## ACKNOWLEDGEMENTS

The authors thank the MUN Field Services team for their logistical support and assistance in the field. We thank staff at the Marine Institute's Centre for Sustainable Aquatic Resources (Terry Bungay and George Legge) for assistance in constructing and testing the camera apparatus. We acknowledge Jonathan Bergshoeff for supporting the work, and Kyle Matheson for assistance with local site information and contacts. The authors wish to thank Kiley Best, Stephanie Green, Brett Howard, Blaine Griffen, and one anonymous reviewer for helpful comments to improve this manuscript.

### Funding

This project was funded by a Marine Environmental Observation Prediction and Response (MEOPAR) Early-Career Faculty Development Grant awarded to Brett Favaro (EC1-BF-MUN). Nicci Zargarpour was supported by an Ocean Industry Student Research Award from the Research and Development Corporation of Newfoundland and Labrador (5404-1914-101). Funding was also provided by the Canadian Centre for Fisheries Innovation (H-2015-06), and the Newfoundland and Labrador Department of Fisheries and Aquaculture (currently, Department of Fisheries and Land Resources) Fisheries Development and Diversification Fund (NH-77836). The funders had no role in study design, data collection and analysis, decision to publish, or preparation of the manuscript.

### Grant Disclosures

The following grant information was disclosed by the authors:
Marine Environmental Observation Prediction and Response (MEOPAR) Early-Career Faculty Development Grant awarded to Brett Favaro: EC1-BF-MUN.
Ocean Industry Student Research Award from the Research and Development Corporation of Newfoundland and Labrador: 5404-1914-101.
Canadian Centre for Fisheries Innovation: H-2015-06.
Newfoundland and Labrador Department of Fisheries and Aquaculture (currently, Department of Fisheries and Land Resources) Fisheries Development and Diversification Fund: NH-77836.

### Competing Interests

The authors declare there are no competing interests.

### Author Contributions

- Nicola Zargarpour conceived and designed the experiments, performed the experiments, analyzed the data, prepared figures and/or tables, authored or reviewed drafts of the paper, and approved the final draft.
- Cynthia H. McKenzie conceived and designed the experiments, authored or reviewed drafts of the paper, and approved the final draft.
- Brett Favaro conceived and designed the experiments, analyzed the data, authored or reviewed drafts of the paper, and approved the final draft.

### Animal Ethics

The following information was supplied relating to ethical approvals (i.e., approving body and any reference numbers):

The project was approved as a 'Category A' study by the Institutional Animal Care Committee at Memorial University as it involved invertebrates (project # 15-02-BF).

### Field Study Permissions

The following information was supplied relating to field study approvals (i.e., approving body and any reference numbers):

All field research was conducted under experimental license NL-3271-16 issued by Fisheries and Oceans Canada.

## Data Availability

Raw data are available in the Supplemental Files.

## Supplemental Information

Supplemental information for this article can be found online at http://dx.doi.org/10.7717/peerj.8444#supplemental-information.

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
