# Peer review of "A field-based investigation of behavioural interactions between invasive green crab (Carcinus maenas), rock crab (Cancer irroratus), and American lobster (Homarus americanus) in southern Newfoundland"

_PeerJ, doi:10.7717/peerj.8444_

## Round 0.1 · original submission · Major Revisions

We have received three reviews for your manuscript. All reviewers concurred that your manuscript could potentially make a valuable contribution to PeerJ. However, they all also agreed that several aspects of the paper deserve major revisions at this stage.

In particular, the manuscript needs a major rewriting effort. The length of the paper needs to be shorten and the number of figures reduced. Importantly, the introduction needs to be restructured and refocussed on the main topic of the paper, with clear research questions and rationale being included. Several parts of the methods and results could also be more concise, and some parts of the discussion need to be cut/streamlined. The general conclusions and their importance also need to be improved.

Some parts of the statistical approach needs to be adjusted and detailed further. For instance, reviewer 1 suggested that high densities could affect the results, and this potential problem should be answered / discussed convincingly in the next version. Furthermore, as suggested by reviewer 2, the effects of depth and body size/claw size on the results should also be addressed/included in the analyses.

All other comments provided by the reviewers also need to be integrated in the next version of the manuscript.

·

Basic reporting

This is an interesting study that uses field experiments and observations to test the hypothesis that the invasive European green crab could prohibit lobster from entering traps and may therefore be responsible for recent declines in lobster catch. The authors conclude that this is indeed possible, but I have substantial reservations about this conclusion. I outline my concerns, as well as other suggestions for the manuscript, below.
1) This is a very long paper. The length could be reduced by removing portions of the Introduction that are not really pertinent. For instance, there is a lot of information on genetics and temperature tolerance of green crabs, on their impacts on eelgrass beds, and on previous eradication methods that, while interesting, is not pertinent to the topic of this paper.
2) The Discussion of the paper is FAR too long! I’m not sure that it is necessary to separately discuss each and every part of the study. Discussing the study as a whole instead would be much more concise.

Minor comments
1) Line 127 – remove “are”
2) Lines 265-269 – description of Poisson GLMM not necessary. This is a standard ecological test for count data.
3) Lines 334-336 – explanation of prop.test is redundant with explanation above
4) Perhaps give the number of traps deployed and the numbers of hours of video footage in the Methods section rather than in the Results.

Experimental design

1) The researchers used video cameras to observe interactions in and around lobster traps, but they simply used ambient light to do this. Given that crustaceans are often more active at night, it may be that many interactions were not observable. What impacts may this have had on the results of the study?
2) Lines 198, 223 – I recognize that the green crab is an invasive species, but collecting individuals for “disposal on shore” seems potentially ethically problematic. How were these individuals disposed of? The authors state that the project was approved by the University Animal Care Committee, but I think it is still worth ensuring that animals were disposed of humanely and weren’t just thrown in a dumpster in a plastic bag to die slowly in the heat.
3) SCUBA surveys – if one diver pulled out the transect line right before the other diver did the survey along the line, it is likely that the activities of the first diver would have disturbed crabs and lobster, causing them to either hide or leave the area. These surveys may therefore have underestimated the abundance of animals. And these artifacts, if they occurred, could have been species-specific.
4) Lines 274-276 – Perhaps a multi-model comparison using AIC would be a more robust method of identifying the best-fitting model.

Validity of the findings

The premise of this study is that green crabs are reducing the catch of lobsters by interfering in some way in or around the traps. The authors conclude that crabs inside traps reduces lobster entrance into traps. However, I have fairly serious concerns with this conclusions for the following reasons.
This study was conducted in an area where lobster fishing had just been closed, and thus the lobster densities were presumably relatively low because of the previous fishing. Yet this study caught a total of 326 lobsters and just 6 green crabs in traps. Further, few free crabs were observed in 452 h of video (line 390-291). This does not provide much evidence that green crabs are entering traps. This suggests that the high density of crabs tethered in the traps for the experimental treatments was much higher than the density of crabs expected to enter lobster traps normally. Thus, the impact of the tethered crab treatments on lobster catch here is likely hugely overestimated.
Further evidence that crabs are causing a problem in lobster traps comes from the fact that there was no impact of crab presence on the success of small lobsters in entering the traps. Interactions between crustaceans are strongly determined by individual size. We should expect therefore that crabs would have the largest impact on the smallest lobsters. The fact that there was no impact on these small individuals suggests that crabs are not the problem here. I think that the findings here that fewer large lobsters entered traps with crabs simply reflects the incredibly (and unrealistically) high density of crabs (7 adult crabs in an area much smaller than 0.25 m2) in a confined area. This will of course lead to an increase in aggressive interactions both between crabs and between lobster and crabs.
Finally, there was no evidence here that crabs depleted bait in the traps, even when they were stocked at these crazy high densities. So the mechanisms was instead likely direct aggressive interference as lobsters entered the traps. But the fact that all predation attempts observed were of lobster eating crabs, further suggests that crabs are not deterring lobster in general (i.e., aggressive interference from crabs at reasonable densities would likely have weak impacts on lobsters). The most likely scenario for a lobster encountering a trap is to have no crabs inside (based on the observation that few crabs entered traps here). The next most likely scenario is to have crabs inside, but at a much lower density than used in this experiment. In this case, results here suggest that the lobster would view the crab as a prey item, not as a problem or a reason not to enter the trap.

·

Basic reporting

I love the premise of this study, as it addresses an urgent question for the management of green crabs on the east coast in a very applied way, however the quality of the writing is not suitable for publication. The introduction in particular needs to be rewritten or significantly restructured so the relevant pieces of information are grouped together, rather than scattered throughout. I am also unsure as to why there is so much focus on phenotypic plasticity and genetics in the intro when it is neither alluded to or directly addressed in the remainder of the MS. There are also places where the authors are not clear about their meaning. I have provided some examples below.

I would suggest that the authors use “green crab” rather than “crab” in the introduction (and throughout), to distinguish between green crab, rock crab, and crabs generally.

Suggest removing ‘tethering’ from the title, as this is not a formal tethering experiment.

1. I suggest re-phrasing the concluding sentence of the abstract (lines 41-43), for clarity. For example: Our results suggest that while there is a relationship between in-trap crab density and trap effectiveness, it is not linked to the non-native / native status of the crab species.
2. Lines 52-53: Specify that you mean adult tolerances here (vs larval)
3. Lines 56-62: Examples of the impacts detected would be helpful here (rather than Lines 90-94). Specify that you mean impacts on the east coast of North America (as impacts elsewhere may be different).
4. Line 65: Specify the native range
5. Lines 101-111: This paragraph is out of place, as the impacts of green crab were already discussed above (so this section should be moved up). Following the preceding impacts, you need to expand on the issues with lobster here, including competitive interactions with green crabs.
6. Lines 127-128: Can you explain what about them isn’t understood in more detail? The preceding paragraph outlined a number of studies that did look at lobster-crab interactions.

The discussion has some similar issues.

1. Lines 440-458: Why is it unclear? It doesn’t seem all that biologically surprising if a large rock crab is already in the trap that you wouldn’t get small lobsters entering. This is potentially very interesting from the perspective of naivete on the part of the lobsters. Are they recognizing a native competitor/predator but not an invasive one?
2. Lines 527-538: From this, I don’t understand how significantly fewer entry attempts by lobsters on green crab stocked traps does not also cause a significantly lower catch of lobsters?
3. Lines 564-566: Do you mean lobster abundances were consistent for all traps for each deployment?

Experimental design

I appreciate the direct and very applied methodological approach used and commend the authors on how many hours of video they scored! I have no fundamental issues with the experimental design but there are two large oversights in the analysis that would need to be addressed before publication: set depth and body/claw size.

Increasing depth, even over a few meters, could result in large changes in the species composition and size of decapods (e.g. larger crabs found at depth). Given the traps are set over a large range of depths, I would like to see depth accounted for in some way, preferably statistically as a co-variate in the models.

Similarly, the body size (and more importantly to interspecific interactions, the claw size) of decapods is crucially important in determining the outcome of agonistic interactions (e.g., Baillie & Grabowski 2018, Ecology). While the authors do give some consideration to “large” vs “small” lobsters, the issue of size is not well explored statistically, despite it being a primary explanation for their results in the discussion. I think there are a number of ways size could be analysed more thoroughly. For example, measuring the claw size (or perhaps meral spread) of decapods interacting on video to see if the larger decapod always wins. This might also provide some information about how tethering may have affected the results.

There are also some clarity issues to resolve in the methods/analysis sections and figures/tables.

1. Lines 174-175: Were these depths reflective of where pots are set to catch commercial/recreational lobster?
2. Line 185: Were depths randomly assigned or haphazard? Are depths standardized against mean low water?
3. Line 199: In the same location?
4. Line 222: Specify whether notch-notch or point-point measurements were used.
5. Line 228-229: Unclear. Three dives were done per deployment, one per pot? Or were all pots surveyed three times per deployment.
6. Line 253: Figure number?
7. Line 298: Negative-binomial?
8. Line 315-316: If the transect was done over 25 m moving away from a baited trap, how are they independent? I would expect the bait to attract crabs and lobsters, leading to higher densities as you move towards the trap?


Figure 5: Suggest the points be made different shapes and smaller.

Figure 6 and 7: Suggest the captions needs to provide more explanation (e.g. that this is from the video data analysis, what is going on for deployments 5/6, how specifically these two figures differ).

12 figures in the main MS is overwhelming. I recommend paring the figures down to those that pertain most directly to your main findings/hypotheses, and creating a supplementary file for the remainder.

What does ‘standard trap’ mean in Table 1? Check for consistency in treatment names throughout figures.

Validity of the findings

If the authors can statistically address the effects of depth and body size/claw size on their results, then I would likely consider their findings valid.

Additional comments

I think the use of video to understand interspecific interactions of these species is a fantastic approach to a really important, applied, question. Beyond its applied aspect however, it is also a great approach to studying the behavioural ecology of invasive and native decapods (as the authors mention, video provides a very fine scale for making observations). However, the authors state several times in the discussion that their findings are 'unclear'. I think using the invasion ecology and behavioural ecology literature more extensively would provide some clarity.

Reviewer 3 ·

Basic reporting

Per comments below, there are some issues with the reporting, especially relating to the need to be more concise and clarify methods and results. Also, the discussion appeared to me to be a really long results section rather than a synthesis of information and how it relates to other published work.

Experimental design

I think the design is fairly sound. I was a little uncertain of why the authors tethered the crabs when they were doing video capture.

Validity of the findings

I think validity is fine. The way presented needs revising per above and below.

Additional comments

General Assessments:
This MS presents a study of the potential impact of green crabs on the capture of lobsters in Newfoundland. Given the crab’s general impact in the region, this represents an important avenue of study, particularly given the decline of lobsters in the region and the potential role that green crabs may play in that decline or lobster recovery. I think this MS is worthy of publication, but there are some issues that need to be resolved prior to publication. First, the introduction has some mischaracterizations and missing information regarding the background information of the green crab invasion in the region (and throughout Nova Scotia). Second, I think it may be better to transform the hypotheses into questions. I found the hypotheses hard to understand at times, and in some cases, I was having trouble following the logic behind them. They also did not always have a clear rationale about why the hypotheses were formulated (one example being a hypothesis for green crabs versus rock crabs affecting catch of lobsters). Third, a lot of the methods and results could be more concise. There was quite a lot of wordiness throughout the manuscript, and it made it difficult to follow at times. I think organizing around your questions/hypotheses is helpful, but the descriptions could be shortened significantly. In line with this, there are too many figures – these should also be condensed. Finally, the discussion to me felt more like a longer version of the results – there was very little return to what was presented in the introduction. For example, to me, there seemed to be arguments made in the introduction related to temperature tolerance in green crabs, but this is never returned to in the discussion. Is there something about that which is of value to this study? Generally, this paper could be written in a way that helps it serve as a contribution to the general understanding of the impacts of green crabs worldwide, but there was very little of that in the discussion – it was very focused on just explaining the findings in this study, rather than how the findings of this study link to other findings and what the implications could be to our understanding of this globally invasive crab.

I have other minor suggestions in the appended .pdf. I also have specific comments about certain sections which are in the appended .pdf but also copied and pasted below.

--Third paragraph of introduction: Missing in here is the discussion of the potential for neutral processes in the green crab invasion between the two lineages... this is discussed in Pringle et al. (2011), Darling et al. (2014), and Lehnert et al. (2018). These should be introduced in here as alternatives (or potentially also at play) to the temperature tolerance hypothesis.
--Introduction 4th paragraph: “These populations contain both southern and northern genotypes and genetic analysis indicates a close match to the more cold-tolerant, northern populations (Blakeslee et al., 2010; Roman, 2006).” This is a misrepresentation of these papers to go along with the hypothesis of temperature tolerance. What the genetic data tell is that the Newfoundland source was from Nova Scotian populations that are associated with shipping between Nova Scotia and Newfoundland. There was nothing about temperature in there and there is no way to even make that conclusion using those data. This sentence needs to be revised for accuracy. Moreover, the Roman paper did not look at the Newfoundland data because there were no green grab populations detected in Newfoundland at the time of the sampling for that paper.
--Methods: Tethering. In general, I'm not really clear on the point of tethering if you were going to video the crabs anyway. It seems to have added an extra variable to have to deal with. Why tether? More explanation is needed here.
--Discussion: “It is unclear why rock-crab pre-stocked traps captured fewer lobster than unstocked traps.” This seems to suggest you had an a priori idea that rock crab stocked traps would inhibit lobsters less, but I don't see any clear reason for that. Are you suggesting lobsters are more likely to avoid green crabs than rock crabs? If so, there needs to be some kind of evidence presented to support that. It may simply be a density dependent effect, regardless of crab identity.

Annotated reviews are not available for download in order to protect the identity of reviewers who chose to remain anonymous.

---

## Round 0.2 · Minor Revisions

We have received three additional reviews on the new version of your manuscript. While some of the revisions were deemed satisfying, others were not. In particular, the writing and presentation of findings still need some work and one of the reviewer made several suggestions that will help further improve this aspect of your work. Additional clarifications/justifications regarding the experimental design and analyses (trap, depth, density, etc.) are also required.

·

Basic reporting

This version of the paper is much improved. The authors have substantially clarified their points and have removed extraneous material in the Introduction. Much appreciated! The Discussion is still excessively long, but it is better.

Experimental design

The experimental design has been clarified, and I'm happy enough with it - though I still contend that AIC is a superior way to select the appropriate model as compared to stepwise model selection techniques.

Validity of the findings

No comment.

·

Basic reporting

A lot of effort has clearly been made and the improvement to the Introduction is particularly noticeable. However, it is my opinion that the writing still needs significant work to effectively communicate the findings of this study. A lot of really interesting and relevant angles remain under- or un-explained. I think this is partly (perhaps largely) due to a need for better organization/flow in the writing.

My specific comments hopefully clarify the areas where I feel the authors need to expand or re-organize, but in general the link between the impact of an invader on a very important commercial fishery deserves more attention, as does the behavioural ecology.

Specific comments and suggestions on the reporting
Review the use of crab vs crabs throughout for consistency. Are you using ‘crab’ as both the singular and collective noun, or crab (singular) and crabs (collective)? Same for lobster/lobsters. For example, there’s a switch between these uses in line 37 where the lobsters (collective) eat crab and lobster (also collective). It is either “the lobsters ate crabs and lobsters” or “the lobster ate crab and lobster”.

Review the order in which methods and results are presented. In most places the green crab treatment is described first, then the rock crab, but this is reversed in places (e.g. line 285).

Clarify the distinction between legal/sublegal vs. adult/juvenile lobster (especially in the Discussion).

23: Move ‘marine’ up to the beginning of the sentence (to distinguish from terrestrial invaders, which probably don’t have an impact on marine biodiversity), could also add ‘native’ in front of biodiversity. However, this paper isn’t about biodiversity really, its about the impacts of an invader on a commercial fishery. That would be a more relevant opening.

25: Newfoundland, Canada

26: Specify that lobster are native to the region

34-27: Combine 1 and 2 for brevity/clarity (“Regardless of the species of crab stocked, crab presence reduced the total number…. compared to the empty control traps.”)

39: I am skeptical of the word “newly” here. What defines a new invasion and why does it matter that it is new? There is no substantive discussion of why the invasion being new is relevant to the findings (here or on the actual Discussion).

46 (paragraph): The assertions made here should be backed up with examples from the literature, to better make the case for linking invasion ecology research to fisheries management. There are interesting examples across taxa where a marine invasive was responsible for the decline of a fishery (e.g. Mnemiopsis ledyi, Indo-Pacific lionfish, etc). The latter would go a long way to making this paper conceptually interesting to a larger invasion ecology audience and not just us green crab nerds!

55: Remove “finally”

60: Remove the apostrophe in 2000s

63: Can be severe (they are not in all cases, eg. South Africa, Australia).

61-71: Why emphasize the eelgrass thing and not their impact on fisheries through direct species interactions (which is what the paper is about)? I would suggest expanding on their impact as a predator and/or competitor. The indirect effect of green crabs on ecosystems via eelgrass loss is important, broadly speaking, and deserves mention but much less relevant than the direct impacts in this context.

77: I think ‘traditionally’ here may undersell the commercial importance of lobster to Nfld?

80: “…presence of the species” instead of invasion here.

83: Most readers won’t understand or know where Fortune Bay is or if/why it matters. Either identify and reference on Figure 1, or generalize the statement (e.g ‘ongoing expansion of the species into still-productive lobster fishing grounds…”).

90: Conclude with something like “However, no studies to date have examined how these species interact outside the lab”. When you think about it, it’s a huge and important step to take what we know from the lab and move it into the field. I think this deserves emphasis, here and in the Discussion.

92: Presumably there have been some studies on lobster recruitment in Newfoundland? The central argument between intraspecific population dynamics of lobsters vs interspecific interactions between lobsters and crabs as the reason for declining catches is rather scattered throughout and the population dynamics part needs more support.

91 (paragraph): I think this paragraph is quite methods-heavy for the intro.

111: I think the “ie.” part here is unnecessary

137: ‘zodiac’ is a brand name (Zodiac) and would likely not be recognized as a colloquialism for the type of boat by many/most readers. Change to rigid-hull inflatable boat (or whatever it was).

144: Specify that this is 50 m horizontally, not vertically.

147: Here it says crabs, but could that not mean rock crabs?

148: In accordance with Memorial University’s animal care protocol?

155-159: While I have no problem with the stocking densities, they should be better justified here for the sake of clarity.

206: Move this sentence up to the end of the preceding paragraph (line 201).

202-208: A reasonable effort was made to address the issue of size differences I put to the authors in the first revision. However, none of that made it into the paper? The method used to determine the size of claws for crabs involved in the 60 predation attempts should be described here, and should also be mentioned in the Results and Discussion.

240-244: This doesn’t need to be re-stated in the Methods as this justification is (or should be) in the Intro already.

Results (overall): I think it should be explicitly stated which fixed effects were dropped and which remained in the final model for each of the GLMMs (only done for SCUBA results?).

Section 3.1: I am confused by the way the information contained here is broken up into paragraphs. The topic sentences don’t match the subsequent content and makes it hard to follow.

312: Clarify – I can’t tell this means “(i.e., excluding….)? As is, it reads as though the authors did count the lobsters entering the sides?

Discussion (overall): In general, there is repetition of the results in the discussion (eg. line 365-366), and there is repetition within the Discussion itself (e.g line 456). It may be worth considering structuring the Discussion based on the hypotheses presented in the Intro, rather than the experiments from the Methods, to reduce repetition?

The whole Discussion still needs work to make the findings and their biological importance clear. As suggested by another reviewer, this paper could slot nicely into the larger green crab impacts literature, especially given its commercial fishery implications, but to do this the authors need to go beyond restating and explaining the findings of the paper in isolation. While this has been improved on this revision, I agree with the reviewer and feel there is a ways to go to fully accomplish this.

372 (paragraph): I feel like there is a better way to present this limitation. There are average or high stocking densities, right? So it is less likely that the results have underestimated the effect.

397: The figures also show this as the catches often/consistently asymptote (Fig 5).

407: Does not need to be re-stated

412: This paragraph is a useful example of some of the things that could be improved from a writing perspective. First, the topic sentence doesn’t really clarify what this paragraph is about as the next sentence is about something completely different and arguably more important. Second, the paragraph is a re-statement of results but there is no discussion provided. For example, if bait depletion has been put forward as an explanation for lower lobster catches demonstrating why that isn’t the case is important! Could there be a biological explanation (is there evidence in the literature that green crabs don’t feed when stocked at density because of competition, or they don’t like the bait provided, etc). Or is it an experimental design problem and, if so, what could be done in future studies to better answer the question of bait depletion?

496-501: This assertion needs to be much more clearly articulated. I draw the opposite conclusion from this. Ie., that there was a failure to observe what was actually a very large population of green crab in the first year on SCUBA.

Experimental design

Sections 2.6 – 2.8: Going back to my previous comments, I am still not wholly satisfied with the treatment of depth in this paper.

BH: Increasing depth, even over a few meters, could result in large changes in the species composition and size of decapods (e.g. larger crabs found at depth). Given the traps are set over a large range of depths, I would like to see depth accounted for in some way, preferably statistically as a co-variate in the models.

Authors: We added clarification of the depth ranges at which traps were deployed (3 to 19 m mean ± 1 S.D. = 9.52 ± 4.27 m , as measured at low tide). Within each deployment, the traps were deployed at depths within a few meters of each other. Therefore, within deployments, differences in depths did not influence our response variable. (Lines 135-136)

The clarification on depth range is useful. However, I don’t think the authors can say that differences in depth did not influence the response variable without better (statistical) justification. Although the deployment number is used as a random effect in the relevant models, that still wouldn’t address within-deployment variation in depth. A few meters change in depth would definitely affect the size and number of decapods caught in other systems, so the authors need to provide evidence that it does not matter in this system. Given that the actual depths are known I can’t see a reason not to do this statistically? Since the authors use stepwise model selection, adding depth as a covariate from the start seems the most straightforward to me. However, maybe simpler models (catch ~ depth and size ~ depth) compared to a null model would work (as a supplement even)?

Validity of the findings

As before, I think the findings are interesting and valid. They did the best they could with the available data to address the issues of size I brought up previously (although that work should go into the paper!). But I feel depth still needs to be addressed.

Additional comments

This paper has a lot of potential, but the presentation of the paper falls short of what I would expect for scientific publication. I realize that this is a high standard, but I believe the science done here is worth the effort.

Reviewer 3 ·

Basic reporting

I think the study is straightforward, fairly well-written, and provides important information on the potential impact of invasive green crabs on lobsters in a newly invaded region of Newfoundland. My major comment is the lack of information on densities of green crabs in Fortune Bay in the manuscript (the authors suggest it is fairly low, and the data on green crab captures in this manuscript also suggests that it is low), and with low crab abundance, one might suspect that the impact of green crabs may be limited (at this point in the invasion). It would have been interesting to compare Fortune Bay to Placentia Bay where green crab abundances (based on past work by DFO and other published studies) are much higher, and where the expectation for an impact on lobster catches and lobster densities may be greater. In the newly invaded location, lobsters may also be more naive to the green crab, so could the lack of a difference between green crabs and rock crabs be related to that naivety? A comparison with an older invasive population could provide some understanding there too. I think it would be helpful to reflect on this -- is there any evidence that naivety to green crabs could be a factor in community responses to its presence in a system?

Altogether, I think these data should be published, but I do think the study is a lot less impactful within a single bay, versus comparisons to other bays where the green crab is more established and where its potential impacts on native species may be expected to be more pronounced.

Experimental design

Per my comment in the .pdf, I think more up front description of the trap (and it's effectiveness) would have been helpful. Also, information on abundances/densities of green crabs and rock crabs in Fortune Bay would be important to include. Otherwise (aside from my comment above about comparing to a more green crab established bay), I think the experimental design is straightforward.

Validity of the findings

The findings all seem valid based on the current study design.

Additional comments

My other major comments is that I think the discussion is too long and rehashes the results a bit too much at times. I would like to see more synthesis of the results and why they are meaningful. I have other minor comments in the appended .pdf.

Annotated reviews are not available for download in order to protect the identity of reviewers who chose to remain anonymous.

---

## Round 0.3 · accepted · Accept

I am generally pleased with the final revisions performed on the manuscript.